# Current Knowledge of Th22 Cell and IL-22 Functions in Infectious Diseases

**DOI:** 10.3390/pathogens12020176

**Published:** 2023-01-23

**Authors:** Kunyu Zhang, Lei Chen, Chenyu Zhu, Meng Zhang, Chaozhao Liang

**Affiliations:** 1Department of Urology, The First Affiliated Hospital of Anhui Medical University, Hefei 230022, China; 2Anhui Province Key Laboratory of Genitourinary Diseases, Anhui Medical University, Hefei 230022, China; 3Institute of Urology, Anhui Medical University, Hefei 230022, China; 4The Second Clinical Medical College, Anhui Medical University, Hefei 230032, China

**Keywords:** Th22 cells, IL-22, immunity, infection

## Abstract

T helper 22 (Th22) cells, a newly defined CD4+ T-cell lineage, are characterized by their distinct cytokine profile, which primarily consists of IL-13, IL-22 and TNF-α. Th22 cells express a wide spectrum of chemokine receptors, such as CCR4, CCR6 and CCR10. The main effector molecule secreted by Th22 cells is IL-22, a member of the IL-10 family, which acts by binding to IL-22R and triggering a complex downstream signaling system. Th22 cells and IL-22 have been found to play variable roles in human immunity. In preventing the progression of infections such as HIV and influenza, Th22/IL-22 exhibited protective anti-inflammatory characteristics, and their deleterious proinflammatory activities have been demonstrated to exacerbate other illnesses, including hepatitis B and *Helicobacter pylori* infection. Herein, we review the current understanding of Th22 cells, including their definition, differentiation and mechanisms, and the effect of Th22/IL-22 on human infectious diseases. According to studies on Th22 cells, Th22/IL-22 may be a promising therapeutic target and an effective treatment strategy for various infections.

## 1. Introduction

Different T lymphocytes express distinct surface antigens, allowing them to be subdivided into CD4+ and CD8+ T cells. As a significant component of innate and adaptive immunity, CD4+ T cells protect individuals from infection. Nevertheless, pathogenic CD4+ T cells in some inflammatory illnesses contribute to disease progression [1]. Before migrating to the peripheral blood, progenitor T cells mature into naïve CD4+ T cells after a series of selections in the thymus. Upon antigen stimulation in peripheral blood or lymphoid organs, activated CD4+ T cells differentiate into different lineages to exert biological effects on diverse pathological processes [2]. According to their functions and different cytokine profiles, CD4+ T cells are further categorized into seven subsets, including T helper 1 (Th1), Th2, Th17, Th9, Th22, regulatory T cells (Tregs) and follicular helper T cells (Tfhs) (Figure 1A) [1,3,4,5]. Moreover, with rapid advances in biotechnology, including CRISPR genome editing, single-cell RNA sequencing and structural biology, novel CD4+ T-cell subsets continue to be identified, which have contributed to a deeper understanding of CD4+ T cells in human diseases [5].

### 1.1. The Discovery of IL-22 and Th22 Cells

Initially termed IL-10-related T-cell-derived inducible factor (IL-TIF) [6], interleukin 22 (IL-22) is an IL-10 family chemokine [7,8]. IL-22 was previously reported to be secreted primarily by Th1 cells [9,10], and further studies on Th17 cells have also shown their ability to secrete IL-22 [11,12]. However, Th1 and Th17 cells secrete only 33% and 11% of the total peripheral IL-22 [13]. In 2009, Trifari et al. described a previously undefined IL-22-producing CD4+ T cell subset that coexpresses C-C motif chemokine receptor 4 (CCR4), CCR6 and CCR10 [13,14]. This novel subset is known as Th22 cells, named after their main effector cytokine, IL-22 [15]. Th22 cells have been shown to be the primary producers of IL-22, and they differ from Th1 and Th17 cells in that they do not secrete interferon-γ (IFN-γ) or IL-17 [13]. Moreover, in addition to CD4+ T cells, IL-22 is derived from other lymphoid cells, including CD8+ T cells, innate lymphocytes (ILCs) and natural killer T cells (NKT) [16]. To better define Th22 cells, Mousset et al. proposed a criterion that included a cytokine-staining assay (IL-13^+^, IL-22^+^, IL-26^+^, TNF-α^+^, IL-4^-^, IL-9^-^, IL-10^-^, IL-17^-^ and IFN-γ^-^), intracellular transcription factors (STAT3^+^ and AhR^+^) and flow cytometry detection (CCR4^+^, CCR6^+^ and CCR10^+^) [17].

### 1.2. Factors in the Formation of Th22 cells and IL-22

The differentiation process of Th22 cells and the production of IL-22 have been confirmed to be regulated by multiple factors, including cells, cytokines and transcription factors (Figure 1B).

Studies have demonstrated that plasmacytoid dendritic cells (pDCs) activate naïve T cells and promote their differentiation into Th22 cells by secreting IL-6 and producing tumor necrosis factor-α (TNF-α) [13]. IL-6, together with TNF-α, promotes Th22 differentiation more effectively than IL-6 alone [13]. In addition, conventional DCs (cDCs), Langerhans cells (LCs) and dermal DCs are able to induce the development of Th22 cells as well; pDCs and LCs are particularly effective [13,18]. During the formation of Th22 cells, additional 1,25(OH)_2_D_3_ and pDCs were shown to act together to induce the expression of CCR6 and CCR10 [13]. 1,25(OH)_2_D_3_ was also shown to promote IL-22 production [19]. Nevertheless, Lopez et al. proposed a different view: that 1,25(OH)_2_D_3_ downregulates IL-22 mRNA expression by directly acting on the inhibitory vitamin D3 response zone of the IL-22 promoter gene [20]. In addition, microRNA-31 (miR-31) can facilitate Th22 cell differentiation by suppressing the BTB domain and CNC homolog 2 (Bach2) [21]. The Notch signaling pathway increases the prevalence of IL-22 and Th22 cells, and the hairy and enhancer of split 1 (HES-1) gene is key to this regulatory effect [22,23]. IL-26 enhances IL-22 secretion and Th22 cell proliferation by promoting Ki-67 gene expression in CD4+ T cells [24]. Furthermore, runt-related transcription factor 3 (RUNX3) is a susceptible gene for psoriasis, and its overexpression promotes Th22 cell secretion of IL-22 [25]. Consequently, these discoveries showed that genetic factors can directly regulate Th22 differentiation.

Except for dendritic cells, when cultured with Th17 cells, activated B cells promote IL-22 secretion and the differentiation of Th22 cells by producing TNF-α and activating mechanistic target of rapamycin (mTOR) signaling in naïve T cells [26]. IL-21 alone or IL-21 in combination with IL-23 and IL-1β also contributes to IL-22 production and Th22 cell formation [27]. IL-21 stimulates signal transducer and activator of transcription 3 (STAT3), which then activates the transcription factors aryl hydrocarbon receptor (AhR) and retinoid-related orphan receptor-γt (RORγt) [27]. Trifari et al. discovered that both RORγt and AhR are indispensable in IL-22 production [14]. AhR agonists also facilitate the conversion of naïve T cells into Th22 cells [14]. Notably, in comparison with the controls, the dramatically increased IL-22 frequency in Tbx21-deficient mice demonstrated that the transcription factor T-bet inhibits IL-22 production [28]. T-bet also negatively mediates the identity and role of Th22 cells [29]. Moreover, transforming growth factor-β (TGF-β) has been proven to suppress Th22 cell differentiation [13].

Therefore, in the physiological state, the coregulation of diverse factors contributes to the formation of Th22 cells and IL-22. Furthermore, Plank et al. determined an optimal Th22 differentiation condition for IL-1β, IL-6, IL-23, the TGF-βR inhibitor galunisertib and the AhR agonist (6-formylindole) [3,2-b] carbazole (FICZ) [28]. Under this condition, IL-22 can be effectively produced without contamination by IL-17A [28].

### 1.3. The Effects of Th22/IL-22

As the major effector chemokine secreted by Th22 cells, IL-22 functions through recognition of the IL-22 receptor (IL-22R), which consists of IL-10 receptor 2 (IL-10R2) and IL-22 receptor 1 (IL-22R1) [30]. The combination of IL-22 and IL-22R1 causes structural changes in IL-22, providing a binding site for IL-10R2 [31,32]. Then, the interactions between IL-10R2 and the IL-22/IL-22R1 complex form an IL-22/IL-22R1/IL-10R2 triad and activate downstream signaling pathways [33]. Notably, all tissues detected express IL-10R2, but IL-22R1 is exclusively expressed in nonhematopoietic organs such as the colon, liver and the pancreas especially [34]. IL-22R1 expression was not discovered in immune organs [34]. Therefore, IL-22 does not act on immune cells directly, but it has been proven to modulate immunity through its downstream signal transduction system (Figure 1D) [35].

According to previous studies [36], the effects of IL-22 on host immune defense can be summarized in the following three aspects. (1) Mediating innate immunity. During *C. rodentium* infection, IL-22 induces colonic epithelial cells to release the antibacterial proteins RegIIIβ and RegIIIγ [37]. Other antimicrobial proteins, such as β-defensin 2, psoriasin (S100A7) and calgranulin A (S100A8), have also been shown to be upregulated by IL-22 [34,38]. IL-22 can mediate innate immunity by upregulating cell fluidity [38,39], mucus secretion and the release of other cytokines [11,40]. (2) Protecting the epithelial barrier and regulating cell proliferation. IL-22 can promote the production of tight junction proteins between epithelial cells to enhance the mucosal barrier [41]. IL-22-induced STAT3 activation also maintains intestinal homeostasis by regulating wound healing of the intestinal mucosa [42]. Additionally, when IL-17A is absent, IL-22 can inhibit epithelial cell apoptosis and promote epithelial proliferation and regeneration, exerting tissue-protective effects [43,44]. IL-22 also activates the PI3K-Akt-mTOR pathway to inhibit pulmonary apoptosis and modulate the proliferation of normal human epidermal keratinocytes (NHEKs) and fibroblast-like synoviocytes (FLSs) [45,46]. In addition, IL-22 can induce liver regeneration through suppressor cytokine signaling 3 (SOCS3), a STAT3-targeted gene [47,48]. (3) Disrupting immune defenses through proinflammatory effects. During respiratory inflammation caused by Gram-negative bacteria [44] and bleomycin [43], IL-22 synergizes with IL-17A to recruit immune cells, resulting in extensive local inflammation. Moreover, proinflammatory acute-phase proteins can be released in human hepatoma cells after the injection of IL-22 [49]. However, in diseases such as AIDS, the acute phase proteins mediated by IL-22 are protective [50]. Therefore, the ultimate effects of IL-22 are influenced by multiple factors, including the type of pathogen infected. Notably, IL-22 binding protein (IL-22BP), a natural IL-22 antagonist, can bind to IL-22 and block IL-22R recognition [51,52,53]. Previous studies have revealed that additional IL-22BP neutralizes the effects of IL-22 and reverses the prognosis of diseases, making IL-22/IL-22BP a promising target for disease intervention [51,54,55].

Studies have shown that IL-22 and Th22 cells broadly influence the onset and progression of various diseases. As the role of Th22 cells in autoimmune diseases has been previously described [35], based on the available findings, this review is dedicated to summarizing the effects that Th22 cells and IL-22 exert on infectious diseases, including viral and bacterial infections. Research advances in Th22-targeted therapies for infection are also discussed.

## 2. Th22 Cells in Infectious Diseases

### 2.1. Th22 Cells in Viral Infections

#### 2.1.1. COVID-19

COVID-19 is pneumonia induced by severe acute respiratory syndrome coronavirus 2 (SARS-CoV-2). Studies in the context of infectious diseases have focused on the pathogenic mechanisms of COVID-19 since its outbreak in 2019. Hoffmann et al. proposed that COVID-19 shares some similar symptoms with influenza and respiratory syncytial virus (RSV)-induced pneumonia [56]. According to previous studies, IL-22/Th22 is protective against influenza and RSV pneumonia [54,57,58] and may exert a similar effect against COVID-19. Among patients with fulminant COVID-19-related myocarditis, some met the criteria for multisystem inflammatory syndrome (MIS-A^+^), whereas the rest did not (MIS-A^-^) [59]. Compared to the MIS-A^-^ group, MIS-A^+^ patients showed higher expression of IL-22, a better prognosis and lower mortality [59]. These results suggest that IL-22 may have a protective and antiviral effect in MIS-A^+^ COVID-19 patients. A novel study noted that abnormal dynamic IL-22R1 expression on blood myeloid cells and CD4+ T cells is a characteristic of SARS-CoV-2 infection [60]. IL-22R1 expression on myeloid cells is discriminative for the severity of COVID-19 [60]. However, COVID-19 patients with different prognoses have similar IL-22 levels, suggesting that IL-22 does not affect the outcomes of SARS-CoV-2 infection [61]. Furthermore, the number of IL-22R1-expressing myeloid cells is correlated with the plasma levels of COVID-19-related immune mediators [60]. During the acute phase of COVID-19, the immune response leads to a dramatic increase in several cytokines, including IL-22, which is mainly produced by Th22 cells [62,63]. This process is called cytokine release syndrome (CRS) and contributes to fatal complications, such as multiple organ failure and acute respiratory distress syndrome [64,65]. This result indicated that IL-22R1^+^ myeloid cells may participate in the cascade, leading to CRS and promoting the deterioration of COVID-19. This study also suggested that the IL-22-induced signaling pathway switches from protective to pathogenic as the disease progresses [60]. Therefore, IL-22/Th22 cells may play a critical role in the pathological process of COVID-19, but the detailed mechanism still awaits further research.

#### 2.1.2. AIDS

Acquired immunodeficiency syndrome (AIDS), an infectious illness that arises from infection with human immunodeficiency virus (HIV), is known for its high mortality and prolonged course [66]. In comparison with healthy controls and HIV-infected patients, more acute-phase serum amyloid A (A-SAA) and IL-22 are produced in HIV-exposed but uninfected individuals (EUs) [67,68]. IL-22 has been confirmed to promote the expression of A-SAA in epithelial cells and liver cells [47,49,69]. Moreover, Misse’ et al. coincubated A-SAA with immature DCs in vitro for further exploration. A-SAA is an agonist of the formyl peptide receptor (FPR) and enhances FPR expression on DCs [67]. The FPR promoted the phosphorylation of CCR5 and decreased the expression of CCR5 on DCs, resulting in decreased susceptibility of DCs to HIV [67]. Therefore, it was suggested that the high resistance of EUs to HIV may be associated with IL-22-induced A-SAA.

However, the higher sensitivity of Th22 cells to HIV compared to other CD4+ T cell subsets is attributed to the high expression of CCR5 and α4β7 on Th22 cells [70]. Both CCR5 and α4β7 are the main binding sites for HIV. Therefore, the counts and functional scores of Th22 cells are decreased significantly in the sigmoid colon of HIV patients [71,72]. Moreover, since Th22/IL-22 limits the translocation of commensal bacteria into the systemic circulation [73], Th22 depletion in HIV patients not only causes impairment of the intestinal barrier but also moves intestinal bacteria to the lamina propria [71,72]. The translocated intestinal bacteria eventually enter the circulation and cause systemic immune activation, which is key to the pathogenesis of HIV [72]. Furthermore, viral load in HIV-infected individuals is negatively associated with the level of serum IL-22, and one study indicated that IL-22 may inhibit HIV replication by regulating C-reactive protein (CRP) and IL-10 [74]. Compared to HIV patients with inflammatory disease coinfection, patients without inflammatory diseases showed lower neutrophil activation and a surplus of IL-22 expression [75]. This result suggests that IL-22 controls HIV-related inflammatory injury by regulating neutrophil hyperactivation [75]. In patients infected with HIV-2, a spontaneously attenuating HIV strain, the expression of CCL20 and CCL28 was significantly increased in the sigmoid colon, leading to local recruitment of Th22 cells [76]. Therefore, Th22 cells counteract intestinal CD4+ T-cell exhaustion and maintain intestinal mucosal integrity. This finding may explain why HIV-2 infection is less dangerous than HIV-1 infection. In addition, mucosa-associated constant T-cell (MAIT) levels were positively correlated with Th22 cell frequency in HIV-infected children [77]. Since innate immune responses are effectively regulated by MAITs [78,79], Th22 cells may also be antiviral in HIV-infected children by mediating the level of MAITs. Moreover, compared to HIV-infected individuals with immune responses (IRs), patients without immune responses (INRs) expressed more IL-22 in the colon and showed more severe mucosal damage [80]. After receiving antiretroviral therapy (ART), INRs had poorer immune recovery than IRs [80], indicating that the dysregulation of IL-22 frequency may be responsible for the poor prognosis of INRs. Hence, although Th22 cells are downregulated in HIV patients, they modulate mucosal immunity and acute phase protein expression to suppress the progression of AIDS.

#### 2.1.3. Hepatitis

Viral hepatitis is typically caused by infection with hepatitis B or C viruses [81,82]. Both hepatitis B and C can increase the risk of cirrhosis and hepatocellular carcinoma [81,83]. According to previous studies, IL-22 showed weak anti-HBV effects and no direct antiviral activity against hepatitis C virus (HCV) [84]. Surprisingly, IL-22^-/-^ mice showed increased susceptibility to inflammatory liver damage, demonstrating that IL-22 protects hepatocytes through mechanisms other than direct virus killing [85].

In HBV-infected patients, both intrahepatic and serum IL-22 expression levels are significantly elevated [86]. It has been confirmed that through the STAT3 pathway, IL-22 stimulates the proliferation of liver stem/progenitor cells (LPCs), cells participating in liver inflammatory responses [86]. Moreover, IL-22 production in hepatitis B patients is positively related to the HBV load [87]. By increasing CXCL9 and CXCL10 expression on hepatic stellate cells (HSCs), IL-22 also promotes inflammatory cell accumulation in the liver, leading to increased liver damage and hepatic fibrosis [88]. As a result, IL-22 may be both proinflammatory and profibrogenic in hepatitis B. In addition, elevated circulating levels of Th22 correlated with the severity of HBV-associated acute–chronic liver failure (HBV-ACLF), suggesting that Th22 cells are a negative predictor of prognosis in HBV-ACLF [89]. Nevertheless, according to Kong et al., IL-22 can activate STAT3 and p53 to induce senescence in HSCs [90]. IL-22 levels also negatively correlate with the development of liver fibrosis [91]. Therefore, the proinflammatory or protective effect of IL-22 during HBV infection may vary according to different disease states [92,93].

For hepatitis C, IL-22 mRNA and Th22 cell levels were elevated in the livers of patients with chronic hepatitis C (CHC) compared to controls [84,93,94], indicating that Th22 cells may be recruited to the liver by intrahepatic chemokines [94]. Furthermore, IL-22 induces the proliferation of HSCs, and the Th22 cell level is positively related to the progression of CHC to cirrhosis, suggesting that Th22/IL-22 facilitates HCV-related liver fibrosis [95,96]. However, an in vitro study confirmed that two variants of the gene encoding IL-22BP are correlated with HCV-mediated liver fibrosis and cirrhosis. It has also been proven that high production of IL-22 is correlated with protective immune responses to hepatitis C [51]. Furthermore, during the progression of hepatitis C to cirrhosis, the ratio of IL-22BP/IL-22 increases with the stage of liver fibrosis and peaks at the time of cirrhosis [97]. Hence, the administration of IL-22BP inhibitors, such as IL-18, prostaglandin E2 (PGE2), NLRP3 and NLRP6 inflammasomes [98,99,100], may be a promising therapy for liver fibrosis [51]. This conclusion was questioned by Wu et al. [101]. They stated that the correlation between IL-22 and protective responses in vitro may not be available in vivo because physiological IL-22 levels in the liver cannot be accurately measured in vitro [101], which means that the exact role of IL-22 in hepatitis C remains controversial.

#### 2.1.4. Influenza

Influenza is a highly infectious disease that primarily invades the respiratory system. By attacking lung epithelial cells, influenza viruses can destroy the pulmonary epithelial barrier and lead to abnormal gas exchange and pulmonary effusion [102]. After infection with influenza A virus (IAV), IL-22 levels in the lung tissues of patients are significantly increased and can gradually return to normal as the disease improves [103,104,105]. Moreover, the lung injury caused by H1N1 IAV is more severe in IL-22^-/-^ mice than in controls [57]. These findings suggest that IL-22 may have a protective role during influenza infection. A study showed that some IAV-infected mice can inhibit IL-22 production by producing type I interferons (I-IFN), and they were at a higher risk of secondary infection with *S. aureus* [106]. In mice infected with IAV, IL-22 protects the integrity of the epithelial barrier and inhibits secondary infection by inducing antimicrobial peptides and intercellular junction proteins expressed in the respiratory epithelium [107]. Tight junction proteins promote fluid efflux to alleviate pulmonary effusion in influenza A patients [108,109]. In addition, IL-22 has also been proven to inhibit IAV-induced lung epithelial cell necrosis, suppress inflammatory responses and promote bronchial epithelial cell regeneration [104,105,110]. Since IL-22 can be blocked by IL-22BP, IL-22BP inhibitors have been assumed to be effective at improving influenza prognosis [54]. Nevertheless, in contrast to the upregulated IL-22 expression in patients with a mild infection, the IL-22 levels were decreased remarkably in patients with severe influenza A [111]. Accordingly, IL-22 protects the respiratory system in IAV-infected patients and can be dysregulated with disease progression.

#### 2.1.5. Acute Viral Myocarditis

Acute viral myocarditis (AVMC) is nonspecific interstitial myocardial inflammation. Coxsackievirus B3 (CVB3) infection is the major cause [112]. Compared to the controls, circulating Th22 cells and IL-22 were significantly upregulated in mice infected with CVB3 [113]. Furthermore, in CVB3-infected mice, the anti-IL-22 antibody reduced the antiviral IFN-γ production while increasing the levels of proinflammatory cytokines such as IL-17, IL-6 and TNF-α [113]. Consequently, the antibody resulted in the deterioration of AVMC, demonstrating that Th22/IL-22 can regulate the expression of cytokines and improve antiviral activity and prognosis during CVB3 infection. If left untreated, AVMC can progress to dilated cardiomyopathy (DCM) [112]. Using animal models of CVB3-induced chronic myocarditis and DCM, studies have revealed that Th22/IL-22 also has a protective role in chronic viral myocarditis and that IL-22 can inhibit cardiac fibrosis [114]. Therefore, Th22/IL-22 may be a promising target for treating coxsackievirus-induced acute viral myocarditis, chronic viral myocarditis, and DCM. Nevertheless, in IL-17A^-/-^ mice with AVMC, IL-22 neutralization contributes to improving acute myocarditis while increasing viral replication at the same time [115]. This result suggests that when IL-17A is absent, IL-22 can exacerbate the progression of AVMC and inhibit CVB3 replication.

#### 2.1.6. Other Viral Infections

Hand, foot, and mouth disease (HFMD) occurs mainly in children infected with coxsackievirus A16 (CV-A16) or enterovirus 71 (EV-71), leading to characteristic herpes on the hand, foot, mouth, and buttock [116]. In the acute phase of EV-71-induced HFMD, levels of both circulating Th22 cells and IL-22 are higher than in the convective phase. In addition, HFMD patients complicated with viral encephalitis have higher levels of IL-22 than patients with HFMD alone [117]. This result suggests that Th22/IL-22 may be crucial in the progression of HFMD [118]. Strong bronchitis in infants [119,120] and pneumonia in elderly or low-resistance patients [121] are often caused by respiratory syncytial virus (RSV) infection. The robust Th22 response during the acute phase of RSV infection predicts prolonged hospitalization, indicating a negative effect of Th22/IL-22 on RSV infection [122]. Nevertheless, contrary to the pathogenic effect of endogenous IL-22, exogenous IL-22 activates the STAT3 pathway in RSV-infected cells to promote apoptosis and inhibit viral replication [58]. Therefore, the administration of IL-22 may be effective in RSV therapy. Warts are a contagious skin disease primarily caused by human papillomavirus (HPV) [123]. In comparison with healthy controls, patients with warts produced more IL-22 in the serum, and their IL-22 levels were positively related to the wart counts, suggesting that IL-22 is involved in the immune response against HPV [124]. Yellow fever (YF) is endemic in the tropics and is characterized by acute fever, jaundice, and proteinuria [125,126]. It is typically induced by infection with yellow fever virus (YFV) [125]. Compared with healthy controls, IL-22 production by liver parenchymal cells from YF patients is significantly upregulated [127]. In addition, Mendes et al. speculated that IL-22 participates in M2 macrophage-mediated organ repair and the immune escape mechanism of YFV [127].

Consequently, Th22/IL-22 has both a protective and a proinflammatory role in viral infections, which is greatly affected by factors such as the subtype of virus, the severity of infection and the presence of IL-17A.

### 2.2. Th22 Cells in Bacterial Infections

#### 2.2.1. *Mycobacterium tuberculosis*

*Mycobacterium tuberculosis (MTB)* is a common pathogen that is primarily transmitted via the respiratory tract [128]. Compared to healthy controls, TB patients had lower levels of IL-22 and IL-22^+^ T cells in their plasma [129,130]. Th22 cells are the main IL-22 producers during *MTB* infection, but other subsets, such as Th1 cells, CD8+ T cells, and NKT cells, also secrete IL-22 [131,132,133]. In addition, the bronchoalveolar lavage fluid (BALF) of pulmonary TB patients contains a large amount of IL-22 at significantly higher levels than the corresponding plasma [134]. IL-22 was also found to be abundant in both TB-induced pleural and pericardial effusions [135]. Therefore, it illustrated a possible aggregation of IL-22-producing cells in the disease sites of TB patients. *MTBs* in pulmonary TB granulomas have been shown to recruit IL-22^+^ T cells [136]. During *MTB* infection, since IL-22R is mainly expressed on the surfaces of macrophages in tuberculous granulomas, it may also contribute to Th22 cell accumulation [132,137]. Furthermore, in tuberculous pleurisy, the accumulation of Th22 cells at the disease site was associated with the chemotactic effect of cytokines in tuberculous pleural effusion (TPE) and the pleural mesothelial cell (PMC)-expressed chemokines CCL20, CCL22 and CCL27 [138]. PMCs also promote Th22 proliferation and differentiation by presenting *MTB* antigens [138]. *MTB* infection in humans can result in asymptomatic specific immune responses, latent tuberculosis (TB) or active TB [128]. Bunjun et al. observed a high level of IL-22 in latent *MTB* patients stimulated by *MTB* antigens, and IL-22 accounted for the largest proportion of responsive CD4+ responses [131]. Compared to wild-type controls, IL-22^-/-^ mice showed higher susceptibility to *MTB HN878* and a higher bacterial load in the lungs during the chronic phase of *MTB* infection [132]. In addition, the pulmonary *MTB* load at the early stage is significantly increased in IL-22-deficient mice [139]. Therefore, IL-22 is required in both adaptive and innate immune responses against *MTB*.

Research has shown that in response to stimulation by *MTB*, Th22 cells can evolve into membrane-bound IL-22^+^ (mIL-22^+^) Th22 cells to extend the half-life of IL-22 [137]. More importantly, mIL-22 binds to IL-22R on infected macrophages to inhibit intracellular *MTB* replication [137]. IL-22 also enhances the expression of CCL2 on epithelial cells to stimulate the pulmonary recruitment of macrophages [132]. In *MTB*-infected phagocytes, IL-22 modulates the expression of Rab7 and Rab14 by upregulating the production of calgranulin A [140,141]. Rab7 and Rab14 subsequently inhibit intracellular *MTB* replication and promote phagosome maturation and fusion [140,141]. Furthermore, IL-22 was found to stimulate TNF-α production by IL-22R^+^ macrophages [132]. TNF-α can promote macrophage activation and *MTB* control directly. Additionally, the level of IL-22 in tuberculous pericardial effusion is positively correlated with MMP-9, an enzyme capable of degrading the extracellular matrix [135]. Since the peripheral level of MMP-9 is associated with the severity of tuberculosis [142], IL-22 might also regulate MMP-9 expression to control *MTB* infection [135]. Moreover, IL-22 can promote the proliferation and recovery of pleural mesothelial cells (PMCs) and the closure of PMC layers [138], which facilitates protection of PMCs from tuberculosis-induced damage. Antimicrobial proteins induced by IL-22 in the chronic phase, such as RegIIIγ, Lcn2 and calgranulin A, ensure the structural and functional integrity of the pulmonary epithelial barrier [132]. IL-22 is also involved in the formation of protective lymphoid follicular structures, called inducible bronchus-associated lymphoid tissues (iBALT), through the CXCR5–CXCL13 axis [139]. In patients infected with *multidrug-resistant MTB (MDR-TB)*, decreased Th22 cell responses are associated with high sputum bacterial loads and severe lung lesions, suggesting that Th22 cells influence the antimicrobial capacity of TB patients [129]. Moreover, fewer T cells that release PD-1 or CD57 were found in multidrug-resistant TB patients with high levels of Th22 cells [129]. CD57 and programmed cell death 1 (PD-1) are markers of cellular senescence, so this result suggests that the Th22-induced anti-tuberculosis immune response may also be related to decreased T-cell senescence. Therefore, Th22 cells and IL-22 inhibit the progression of TB by protecting the pulmonary epithelial barrier and regulating *MTB*-specific immune responses.

#### 2.2.2. *Citrobacter Rodentium*

Since *Citrobacter rodentium*-infected mice closely resemble human infectious colitis, they are commonly used in colitis-associated scientific research [143]. *C. rodentium* infection in mice dramatically induces the expression of IL-22 in the intestinal mucosa [143]. Studies have confirmed that during *C. rodentium* infection, early IL-22 production is primarily dependent on ILC3s, whereas IL-22 derived from Th22 cells predominates at a later stage [144]. Th17 cells are also involved in IL-22 production [144]. Further experiments injected purified Th22 cells, Th17 cells or ILC3s into *C. rodentium*-infected mice deficient in IL-22. Compared to the other two subsets, mice injected with Th22 cells had a higher survival rate, reflecting that Th22/IL-22 is critical for the host’s immune response to *C. rodentium* [144]. Moreover, when the STAT3 gene was mutated, *C. rodentium*-infected mice showed a loss of Th22-induced immune defense, including impairment of the intestinal epithelial barrier, reduction of antimicrobial peptides, and increased dissemination of pathogenic bacteria [143]. This confirms that STAT3 is indispensable for the protective role of IL-22 during *C. rodentium* infection. In addition to STAT3, Th22/IL-22-related effects are dependent on T-bet and AhR [143,144]. Furthermore, IL-22 can induce colonic epithelial cells to express the antibacterial proteins RegIIIβ and RegIIIγ, both of which play a direct bactericidal role in *C. rodentium* infection [37]. Researchers have shown that antibiotic treatment reduces RegIIIγ production in mice, but oral administration of lipopolysaccharide (LPS) can restore it [145]. This indicates that LPS, the principal ingredient of the Gram-negative bacterial outer membrane, is key to the expression of RegIIIγ [145]. The addition of the Toll-like receptor (TLR)-5 agonist flagellin or the binding between TLR and its ligand MyD88 has been revealed to initiate the production of IL-23 by DCs and promote IL-23 to upregulate IL-22 production and RegIIIγ expression [146,147]. As a consequence, the host defense against *C. rodentium* induced by Th22 cells is dependent on STAT3 activation and antibacterial protein expression.

#### 2.2.3. *Streptococcus pneumoniae*

*Streptococcus pneumoniae* is a Gram-positive bacterium that colonizes the nasopharynx and initiates respiratory inflammation [148]. According to previous studies, IL-22 expression was rapidly upregulated in the lungs of mice suffering from pneumococcal pneumonia [149]. Moreover, the lung bacterial load was significantly higher in mice with hepatic IL-22R1 deficiency than in controls [149]. Hence, the IL-22-induced downstream cascade is critical for inhibiting *S. pneumoniae* replication. IL-22 was confirmed to promote the clearance of *S. pneumoniae* by upregulating complement factor C3’s expression in the liver and strengthening the binding of C3 and *S. pneumoniae* in the serum [149]. In addition, mice deficient in IL-22RA2 (the gene encoding IL-22BP) showed reduced susceptibility to *S. pneumoniae* and prolonged survival after infection [55]. Further analysis revealed that in the lungs of IL-22RA2^-/-^ mice, oxidative phosphorylation-related genes were reduced in expression. Downregulated oxidative phosphorylation led to an increase in glycolysis and promoted a shift in macrophage phenotypes toward a proinflammatory phenotype, which ultimately upregulated host resistance to pneumococcal pneumonia. Consequently, IL-22 and IL-22BP coregulate antimicrobial immunity against *S. pneumoniae*, suggesting that IL-22RA2 may be an effective target for the treatment of pneumococcal pneumonia [55].

#### 2.2.4. *Helicobacter pylori*

*Helicobacter pylori* is Gram-negative and colonizes the gastric mucosa, causing persistent gastric inflammation, peptic ulcers or even gastric cancer [150,151]. Zhuang et al. demonstrated that Th22 cells are proinflammatory in *H. pylori*-triggered gastritis [152]. Upon *H. pylori* infection, IL-23-induced Th22 cells are rapidly enriched in the gastric mucosa and secrete IL-22, which enhances CXCL2 production by gastric epithelial cells [152]. CXCL2 subsequently bound to its corresponding receptors and resulted in the migration of myeloid-derived suppressor cells (MDSCs) toward the gastric epithelium. In response to IL-22 induction, MDSCs produce the proinflammatory factors calgranulin A (S100A8) and S100A9 and directly inhibit the development of Th1 cells, leading to gastritis progression [152]. Moreover, through activation of the ERK pathway, IL-22 and Helicobacter pylori synergistically promote the production of matrix metalloproteinases (MMPs), particularly MMP-10, in gastric epithelial cells [153]. MMP-10 can aggravate bacterial colonization by inhibiting the production of antimicrobial peptides in the gastric mucosa. In addition, MMP-10 also induces gastric epithelial cells to secrete the chemokine CXCL16, which recruits CD8+ T cells to the gastric mucosa and exacerbates the inflammatory response. However, when IL-22 acts synergistically with IL-17A, it stimulates antimicrobial peptide expression that protects the body from *H. pylori* infection [154]. Moreover, it has been proposed that due to the compensatory effect of other cytokines, IL-22 deficiency does not affect the susceptibility of mice to *H. pylori* or their mortality after infection [154], which contradicts the previous conclusion. According to a paper published in 2019, the discrepancy may be attributed to their choice of different mouse species and *H. pylori* strains, which closely influence the observed immune response [150]. Therefore, the role of IL-22/Th22 in *H. pylori* infection remains to be further discussed.

#### 2.2.5. *Other Bacterial Infections*

*Clostridium difficile* typically causes pseudomembranous colitis or bacterial diarrhea in immunocompromised individuals by facilitating the translocation of enteropathogenic bacteria [155,156]. Compared to the wild-type group, IL-22-deficient mice showed increased mortality after infection with *C. difficile* [156]. During *C. difficile* infection, induced IL-22 upregulates the expression of acute-phase proteins and C3 in the liver and intestine [156]. C3 can be deposited on the surface of enteropathogenic bacteria and enhance the bactericidal activity of neutrophils [156]. Moreover, IL-22-induced glycosylation of host N-linked glycans can promote the growth of *Phascolarctobacterium spp.* [157]. *Phascolarctobacterium spp.* is a succinate-consuming commensal bacterium that competes with *C. difficile* for energy, preventing the growth and colonization of *C. difficile* in the intestine. Infection with *Klebsiella pneumoniae* is often associated with healthcare-related pneumonia or sepsis [158]. Compared with uninfected mice, IL-22 levels in the lungs of mice infected with *K. pneumoniae* were upregulated [44]. The upregulation of IL-22 can reverse the decreased epithelial barrier stability and deteriorated pulmonary inflammation caused by IFN-λ [159]. IL-22 also induces the expression of defense genes such as lipocalin 2 to suppress *K. pneumoniae* [160]. Lipocalin 2 can sequester iron from bacteria to limit their growth [160,161]. Moreover, IL-22 stimulates the lung tissues of *K. pneumoniae*-infected mice to express CCL17 and CCL20 [44], which are ligands of CCR4 and CCR6 [17], leading to the accumulation of Th22 cells in the lungs. *Pseudomonas aeruginosa* is an opportunistic bacterium that can cause an overwhelming local immune response and consistently lead to acute respiratory distress syndrome (ARDS) [162]. The level of IL-22 was transiently increased in *P. aeruginosa*-infected mice, and additional administration of IL-22 reduced their pulmonary damage by downregulating local neutrophil infiltration [163]. IL-22 also induces pulmonary IFN-λ production, preventing the release of inflammatory mediators such as IL-1β [164]. Notably, both protease IV secreted by *P. aeruginosa* [165] and serine protease-3 secreted by neutrophils [166] can degrade IL-22. Therefore, they facilitate the immune escape of *P. aeruginosa* and can result in pulmonary bacterial colonization and the continued deterioration of respiratory function. Consequently, Th22/IL-22 is protective during *C. difficile*, *K. pneumoniae* and *P. aeruginosa* infections. However, *Salmonella enterica serotype Typhimurium* colonizes the gastrointestinal mucosa and typically causes inflammatory diarrhea [167]. The iroBCDE iroN gene cluster [168] and zinc transporter (ZnuABC) [169] in *S. Typhimurium* enable them to escape from IL-22-mediated metal chelation, called nutritional immunity [170]. Since IL-22 is primarily secreted by ILC3s rather than T cells during *S. Typhimurium* infection [171], more detailed mechanisms will not be discussed in this review.

Therefore, Th22/IL-22 exerts a crucial effect on regulating the immune response to viral and bacterial infections. Based on the above, cytokines such as TNF-α, IFN-λ, IL-10 and IL-23; transcription factors such as T-bet and AhR; and chemokines such as CCL2, CCL17 and CXCL13 have been shown to have a protective role in the Th22/IL-22 axis. Other effector molecules, including C3, CRP, MMP-9, Rab7, Rab14, the TLR-5 agonist flagellin and the antibacterial proteins RegIIIβ and RegIIIγ, are also involved in IL-22-induced anti-infectious immunity. Moreover, the matrix metalloproteinase MMP-10; the antimicrobial protein S100A9; and chemokines such as CXCL9, CXCL10, CXCL2 and CXCL16 are risk factors for the Th22/IL-22 downstream signaling system. Interestingly, STAT3 and the antimicrobial protein calgranulin A can be both protective and deleterious in the Th22/IL-22 axis, depending on the type of infectious disease. Notably, IL-22 in infectious illnesses is also derived from ILC3s, Th1 cells, Th17 cells and NKT cells, indicating that the interregulation between different cells ultimately contributes to the development of diseases. This suggests the need for more research on the underlying mechanisms of Th22/IL-22 in infectious diseases.

## 3. Therapeutic Value of IL-22 in Infectious Diseases

### 3.1. Therapeutic Value of IL-22 in Viral Infections

In the treatment of HIV, antiretroviral therapy (ART) has achieved an extraordinary curative effect. After receiving prolonged effective ART, HIV patients exhibited a reconstruction of Th22 cells in the duodenal mucosa lamina propria, which can be attributed to the CCR10-CCL28 chemotactic axis [172]. ART partially restores the immune system of HIV-infected patients. In patients with hepatitis B, studies have found that supplementary IL-22BP can inhibit the growth of LPCs [86]. However, during influenza infection, IL-22BP^-/-^ mice showed a significantly higher survival rate and better epithelial barrier function than wild-type mice [54]. These findings revealed the potential value of IL-22BP in the treatment of hepatitis B and influenza. Furthermore, in the lung tissue of mice with influenza, the novel fusion protein vunakizumab-IL22 (vmab-IL-22), a conjugate of IL-22 with anti-IL-17A antibodies, can inhibit IL-17A-mediated inflammatory responses and elevate the reparative capacity of IL-22 [173]. This study provides a promising direction for drug research in the treatment of influenza. In lung epithelial cells, the novel E3 ligase subunit FBXW12 degrades IL-22R and restricts lung epithelial cell proliferation [174]. It has been proven that FBXW12^-/-^ HeLa cells can increase IL-22R expression and promote cell cycle progression, suggesting that silencing FBXW12 may be a novel therapeutic method for infection-induced epithelial damage [174]. Moreover, in CHC patients, the administration of a γ-secretase inhibitor that suppresses the Notch pathway significantly decreased IL-22 production and antimicrobial responses, suggesting that the modulation of the Notch–Th22 axis may be critical for CHC therapy [175]. Additionally, in mice with ACVM, neutralizing IL-22 can reduce the severity of inflammation and the viral load in the heart when IL-17A is absent [115].

### 3.2. Therapeutic Value of IL-22 in Bacterial Infections

Tripathi et al. found that recombinant IL-22 (rIL-22) can control TB by inhibiting neutrophil infiltration into the alveoli and alleviating lung epithelial cell damage [176]. Moreover, rIL-22 also controls type 2 diabetes (T2D) by lowering insulin and improving serum lipid metabolism. Hence, for T2D patients coinfected with *MTB*, rIL-22 may be a new therapeutic option [176]. In addition, in TB patients, the *MTB* antigens CFP-10 and ESAT-6 can promote PD1 expression in T cells to inhibit IL-22 production [177]. Further experiments demonstrated that the anti-PD1 antibody can restore IL-22 levels and inhibit the development of TB. PstS1 is an *MTB* lipoprotein that promotes the secretion of IL-22 by activating CD8α-DCs in memory T cells, implying that PstS1 is promising for the design of novel tuberculosis vaccines [178]. The TLR ligand MyD88 is indispensable for IL-22 production in immune responses against *C. rodentium* [179] and *S. pneumoniae* [148]. Moreover, the TLR-5 agonist flagellin can upregulate IL-23 levels in *S. pneumoniae*-infected mice. IL-23 then induces IL-22 secretion and triggers an antimicrobial cascade that includes the expression of RegIIIγ [148]. Patients infected with *vancomycin-resistant enterococci (VRE)* are deficient in the intestinal antimicrobial protein RegIIIγ, a condition that disrupts IL-22-induced antimicrobial immunity [145]. Oral administration of either the TLR-4 ligand LPS [145] or the artificial TLR-7 ligand resiquimod (R848) [180] can restore RegIIIγ expression and remodel IL-22-mediated immunity. Therefore, both the selective activation of TLRs and the delivery of exogenous TLR ligands could serve as a novel alternative for treating bacterial infections.

In summary, studies have confirmed the therapeutic effect of Th22/IL-22 on infectious diseases (Table 1 and Table 2). IL-22 aimed at the treatment of viral infections has progressed very rapidly, including the regulation of ART, the IL-22/IL-22BP axis, the Notch-Th22 axis, and even the novel protein FBXW12. Each of them can restore immune homeostasis by regulating the effects of Th22/IL-22. Furthermore, most studies on IL-22-related therapy of bacterial infections highlighted the effects of TLR ligands or agonists, which offered broad prospects for the clinical application of TLR ligands.

## 4. Conclusions

In summary, based on the available studies, we have characterized in detail the roles and mechanisms of Th22 cells and IL-22 in infectious diseases. This finding drives us to conclude that in various infections, Th22/IL-22 can exert a bidirectional protective or pathogenic effect on the human immune system. On the one hand, in most bacterial and viral infections involving epidermal remodeling and mucosal immunity, such as bacterial pneumonia, AIDS and influenza, Th22/IL-22 promotes host defense by mediating the innate immune response and maintaining epithelial barrier integrity. On the other hand, in diseases such as Helicobacter pylori infection, hepatitis B and COVID-19, IL-22 induces disease progression by promoting local recruitment of inflammatory cells and secretion of proinflammatory cytokines. Given the complexity of Th22/IL-22 downstream signaling pathways, further exploration of the role and mechanisms of Th22 cells in infectious diseases is still needed. Our review also summarized recent advances in Th22/IL-22-targeted therapies, which may provide promising insights for infectious disease treatment. Nevertheless, one limitation of the studies we have compiled in this review is that some results regarding the role of Th22/IL-22 in bacterial and viral infections have only been confirmed in mouse experiments, and there is no available human experimental data. Since it is unreasonable to apply the conclusions of animal experiments directly to human beings, there is currently very limited knowledge about how Th22/IL-22 plays a role in human infectious diseases.

## Figures and Tables

**Figure 1 pathogens-12-00176-f001:**
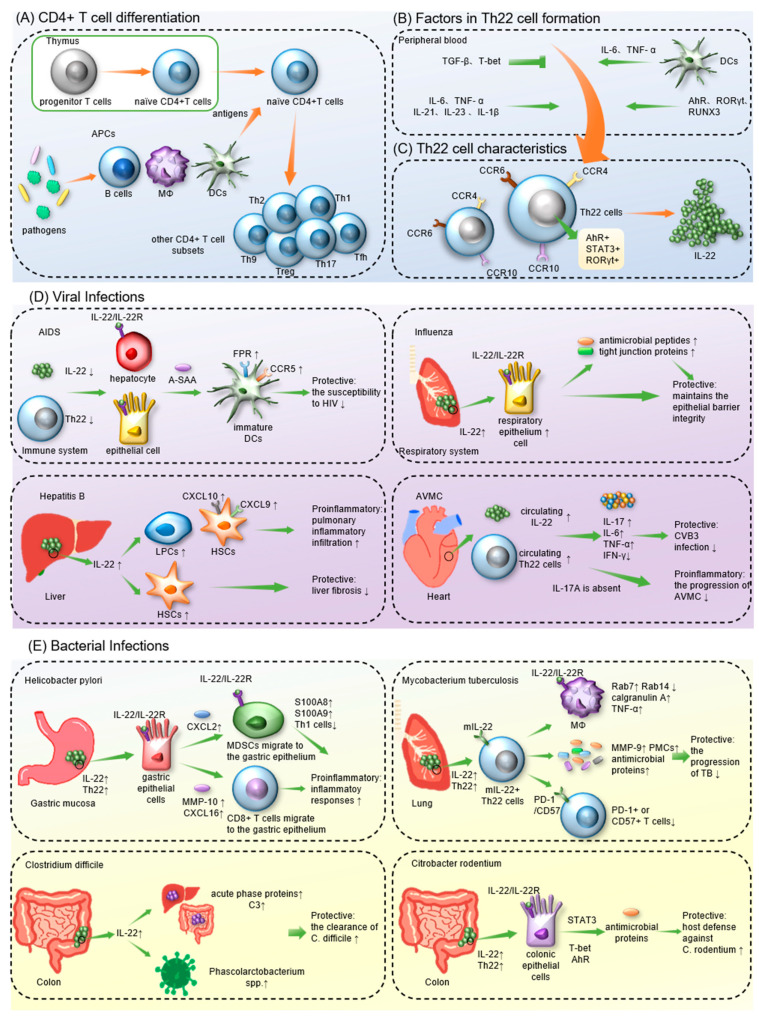
Formation of Th22 cells and IL-22 and the role of Th22/IL-22 in infectious diseases. (**A**). Naïve CD4+ T cell differentiation. In peripheral blood, different cytokines can stimulate mature naïve CD4+ T cells to differentiate into various lineages. (**B**) Factors involved in Th22 cell formation. TNF-α, IL-6 (exogenous or released by DCs) and IL-21; and the transcription factors AhR and RORγt, promote Th22 cell differentiation, whereas TGF-β and T-bet suppress it. (**C**). Characteristics of Th22 cells. Th22 cells are CCR4^+^CCR6^+^CCR10^+^ T cells that express STAT3, AhR and RORγt. IL-22 is the main effector molecule of Th22 cells. (**D**). The roles of Th22/IL-22 in viral infections. (**E**). The effects of Th22/IL-22 on bacterial infections. Th22/IL-22 regulates immune responses during infectious diseases by modulating downstream signaling pathways. APCs, antigen-presenting cells; AhR, aryl hydrocarbon receptor; AVMC, acute viral myocarditis; A-SAA, acute-phase serum amyloid A; AIDS, acquired immunodeficiency syndrome; CCR4, C-C motif chemokine receptor 4; CVB3, coxsackievirus B3; DCs, dendritic cells; FPR, formyl peptide receptor; HSCs, hepatic stellate cells; IFN-γ, interferon-γ; LPCs, liver stem/progenitor cells; MΦ, macrophage; MDSCs, myeloid-derived suppressor cells; PMCs, pleural mesothelial cells; RORγt, retinoid-related orphan receptor-γt; STAT3, signal transducer and activator of transcription3; TNF-α, tumor necrosis factor-α; TGF-β, transforming growth factor-β.

**Table 1 pathogens-12-00176-t001:** Development of IL-22-related therapies.

Infection	IL-22-Related Treatment	Stimulatory/Inhibitory Target	Outcome	References
AIDS	ART	Stimulatory	The percentage of Th22 is nearly restored.	[181]
Hepatitis B	IL-22BP supplementation	Inhibitory	IPCs proliferation is markedly attenuated.	[86]
Hepatitis C	administration of a γ-secretase inhibitor	Inhibitory	Both HCV-reactive Th22 cells and IL-22 levels are reduced.	[175]
Influenza	silencing of FBXW12	Inhibitory	The expression of IL-22R and the progression of cell cycle are both promoted in the lung epithelium.	[174]
	IL-22BP gene knockout	Inhibitory	The survival rate of mice infected with IAV increased.	[54]
	vunakizumab-IL22 administration	Stimulatory	The IL-17A-mediated inflammatory responses are inhibited and the reparative capacity of IL-22 on lung tissue is elevated.	[173]
AVMC	anti-IL-22 antibody used when IL-17A is absent	Inhibitory	The level of Th22 cells and the severity of AVMC are decreased, while virus replication is significantly promoted in the heart.	[115]
TB	recombinant IL-22 used during *MTB* infection complicated with type 2 diabetes	Stimulatory	IL-22 controls TB by suppressing neutrophil infiltration in the alveoli, secreting elastase 2 (ELA2) and inhibiting lung epithelial cell damage. IL-22 also controls diabetes by reducing insulin and improving lipid metabolism in the serum.	[176]
	PD-1 neutralization	Inhibitory	The anti-PD1 antibody restores IL-22 levels and inhibits the development of tuberculosis.	[177]
	PstS1 administration	Stimulatory	PstS1 promoted the secretion of IL-22 by activating CD8α-DCs in memory T cells.	[178]
*S. pneumoniae* infection	TLR-5 agonist flagellin administration	Stimulatory	The activated TLR-5 pathway remodels IL-22-mediated immune responses to defend the bacterial invasion.	[148]
*VRE* infection	oral administration of TLR-4 ligand LPS	Stimulatory	The activated TLR-4 pathway remodels IL-22-mediated immune responses to defend the bacterial invasion.	[145]
	oral administration of artificial TLR-7 ligand resiquimod (R848)	Stimulatory	The activated TLR-7 pathway remodels IL-22-mediated immune responses to defend the bacterial invasion.	[180]

**Table 2 pathogens-12-00176-t002:** Development of therapeutic IL-22/22R blockers.

Therapeutic IL-22/22R Blockers	Types		References
IL-22 cytokine inhibitors	neutralizing antibodies	anti-IL-22 antibodies	[113,115]
	small molecules	T-bet, TGF-β, I-IFN, 1,25(OH)2D3, the *MTB* antigens CFP-10 and ESAT-6	[13,20,28,29,106,177]
IL-22R blockers	small molecules	the novel E3 ligase subunit FBXW12	[174]
	small binding proteins	IL-22BP	[51,52,53]
IL-22BP inhibitors	small molecules	IL-18, prostaglandin E2 (PGE2), NLRP3 and NLRP6 inflammasomes	[98,99,100]

## Data Availability

Not applicable.

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
