# Peer review of "Current Knowledge of Th22 Cell and IL-22 Functions in Infectious Diseases"

_pathogens, 2023, doi:10.3390/pathogens12020176_

Round 1
Reviewer 1 Report
Zhang et al presents an interesting review article surrounding the exquisite crosstalk between Th22 and IL-22 in the context of human infections, and the figure is nicely illustrating and summarizing the goal of this review.
This manuscript is of general interest to the readership of Pathogens Journal
Minor consideration:
The abbreviation Th2 should be spelled out in line 21.
The sentence in line 22 which begin with ( , and the Th22 also express CCR4....) should be a new sentence e.g. Th2 cells express a wide spectrum of chemokine receptors such as; CCR4, CCR6 and CCR 10.
Some typos are present through the manuscript e.g line 174, the sentence should be researches in the context of infectious diseases have focused on....
The language throughout the paper would benefit from a through editing.
Author Response
Point to point reply
Dear Editors and Reviewers:
Thank you for your letter and for the reviewers’ valuable comments concerning our manuscript entitled “Current Knowledge of Th22 Cell and IL-22 Functions in Infectious Diseases” (ID: pathogens-2074797). Those comments are very constructive for revising and improving our manuscript, as well as the important guiding significance to our studies. We have studied the comments carefully and have made extensive corrections which we hope meet with approval. Revised portions are marked under the “Track model” in the paper. The main corrections in the paper and the detailed point-by-point responses to the reviewer’s comments are listed as flowing:
Response to reviewers
Response to Reviewer # 1 :
Comments and Suggestions for Authors
Zhang et al presents an interesting review article surrounding the exquisite crosstalk between Th22 and IL-22 in the context of human infections, and the figure is nicely illustrating and summarizing the goal of this review.
This manuscript is of general interest to the readership of Pathogens Journal
Minor consideration:
Answer:
Dear reviewer,
Thanks very much for your comments!
We have tried our best to revise the manuscript. According to your comments, in the revised manuscript, the full name of Th22 has been added in line 21 as “T helper 22”. Besides, we have been modified the sentence in line 22 into two separate sentences (…, and TNF-α. Th22 cells express a wide spectrum of chemokine receptors such as CCR4, CCR6 and CCR10.). In addition, we have carefully corrected the typos and have asked a professional institution to help us revise the manuscript. Following your comments, our manuscript became more concise and easier to understand.
We hope the revised manuscript meets with approval.
Great thanks again for your appreciation and valuable suggestion!
Best regards.
- The abbreviation Th2 should be spelled out in line 21.
Answer:
Great thanks for your careful checks! We are sorry for our carelessness. The full name of Th22 has been added in line 21, and the revised sentence was listed as below:
Before the revising:
Th22 cells, a newly defined CD4+ T-cell lineage, are characterized by their distinct cytokine profile, which primarily consists of IL-13, IL-22, and TNF-α, and Th22 cells also express CCR4, CCR6, and CCR10 chemokine receptors.
After the revising:
T helper 22 (Th22) cells, a newly defined CD4+ T-cell lineage, are characterized by their distinct cytokine profile, which primarily consists of IL-13, IL-22, and TNF-α. Th22 cells express a wide spectrum of chemokine receptors such as CCR4, CCR6 and CCR10.
- The sentence in line 22 which begin with ( , and the Th22 also express CCR4....) should be a new sentence e.g. Th2 cells express a wide spectrum of chemokine receptors such as; CCR4, CCR6 and CCR 10.
Answer:
Great thanks for your careful checks! According to your advice, the sentence in line 22 has been modified into two separate sentences, which were revised as below:
Before the revising:
Th22 cells, a newly defined CD4+ T-cell lineage, are characterized by their distinct cytokine profile, which primarily consists of IL-13, IL-22, and TNF-α, and Th22 cells also express CCR4, CCR6, and CCR10 chemokine receptors.
After the revising:
T helper 22 (Th22) cells, a newly defined CD4+ T-cell lineage, are characterized by their distinct cytokine profile, which primarily consists of IL-13, IL-22, and TNF-α. Th22 cells express a wide spectrum of chemokine receptors such as CCR4, CCR6 and CCR10.
- Some typos are present through the manuscript e.g line 174, the sentence should be researches in the context of infectious diseases have focused on....
Answer:
Great thanks for your careful checks! We have carefully checked the manuscript and corrected the errors accordingly. The revised typos are listed as below:
“Researchers in infectious diseases have focused on...” in line 174 have been modified into “Studies in the context of infectious diseases have focused on…”; the word “HCV” in line 538 has been modified into “CHC”; in the fourth row of the Table 1, the sentence "the expression of IL-22R and the growth of epithelial are both decreased in lung epithelium." in the fourth column have been corrected into "the expression of IL-22R and the progression of cell cycle are both promoted in lung epithelium".
- The language throughout the paper would benefit from a through editing.
Answer:
Great thanks for your valuable suggestion! According to your advice, we have asked a professional institution to help us revise the manuscript. The revised parts were marked in the manuscript under “Track model”, and some of them were listed as below:
---------------------------------------------------------------------------------------------------------------------
We tried our best to improve the manuscript and made some changes to the manuscript. These changes will not influence the content and framework of the paper. And here we did list the changes under the “Track model” in revised paper. We appreciate for Editors/Reviewers’ warm work earnestly and hope that the correction will meet with approval. Once again, thank you very much for your comments and suggestions.
Correspondence author: Meng Zhang (zhangmeng1930@126.com) & Chaozhao Liang (liang_chaozhao@ahmu.edu.cn).
Reviewer 2 Report
In the submitted manuscript by Kunyu Zhang et al., the authors summarized current knowledge of Th22 cell and IL-22 functions in human infection disease. As IL-22 cytokine and its cognate receptor are generally recognized as “dual-function” molecules playing a substantial role in the tissue regeneration and health-supporting pathways, the same proteins are also known as pro-inflammatory cascade members and, thus, molecular therapeutic targets. The authors focused their attention to description of IL-22 signaling functions in a wide portfolio of infectious diseases. In this context the submitted manuscript brings up-to-date summary of recent research outcomes in this hot topic and fulfils criteria for general interest of the Pathogens journal´ readerships. The manuscript is well prepared, very detailed and its content is adequately supported by the references.
After reviewing the manuscript, I have following comments to the authors:
1. I strongly suggest removing the text super positioning the figure object (text over the drawing) as this loses the visibility and contrast. Please, move text out but in a proximity of the figure object. This is a case of Fig 1 A-B-C-D-E.
2. There are many subjects forming the particular sections of the Figure 1. I will appreciate if the authors include internal frames selecting and separating objects of the Fig. 1A, then Fig.1B a C, Fig. D (each frame for the particular disease) and the same in Fig. 1 E for the particular bacterial infection. This will bring better orientation in the figure and will support the clarity of the figure message.
3. In Figure 1 E, correct the name of pathogens from Mycobacterium Tuberculosis to Mycobacterium tuberculosis, Citrobacter Rodentium to Citrobacter rodentium and Helicobacter Pylori to Helicobacter pylori.
4. While in row 122 the authors use the correct style of citing “Plank et al.”, in many other cases they do not. See row 73 Trifany et al (correct is Trifany et al.), 80 Mouset et al ; 96 Lopez et al ; 113 Trifany et al ; 197 Hoffman et al ; 209 Missé et al ; 343 Mendes et al ; 365 Bunjun et al ; 536 Tripathi et al ;
5. In chapter 2.2.2, correct “Citrobacter Rodentium” to “Citrobacter rodentium”.
6. In Table 1 in “IL-22-related treatment” column, I suggest to clearly indicate whether mentioned item is a stimulator/suppressor and should be blocked/suppressed. For instance FBXW12, IL-22BP, PstS1… It is not clear how it modulators function.
7. In the manuscript, I miss a Table 2 summarizing the development of all important therapeutic IL-22/22R blockers. Such table should contain 3 separated parts: IL-22 cytokine inhibitors, IL-22R1 and IL-22BP blockers. These should be sorted into neutralizing antibodies/antibody fragments (Nanobody); small molecules; and small binding proteins (scaffold binders) (f.i. Affibody, Darpin, Anticalin, etc.). All presented with particular references. This would improve the practical impact of this review article.
Author Response
Thank you for your letter and for the reviewers’ valuable comments concerning our manuscript entitled “Current Knowledge of Th22 Cell and IL-22 Functions in Infectious Diseases” (ID: pathogens-2074797). Those comments are very constructive for revising and improving our manuscript, as well as the important guiding significance to our studies. We have studied the comments carefully and have made extensive corrections which we hope meet with approval. Revised portions are marked under the “Track model” in the paper. The main corrections in the paper and the detailed point-by-point responses to the reviewer’s comments are listed as flowing:
Response to reviewers
Response to Reviewer #2:
Comments and Suggestions for Authors
In the submitted manuscript by Kunyu Zhang et al., the authors summarized current knowledge of Th22 cell and IL-22 functions in human infection disease. As IL-22 cytokine and its cognate receptor are generally recognized as “dual-function” molecules playing a substantial role in the tissue regeneration and health-supporting pathways, the same proteins are also known as pro-inflammatory cascade members and, thus, molecular therapeutic targets. The authors focused their attention to description of IL-22 signaling functions in a wide portfolio of infectious diseases. In this context the submitted manuscript brings up-to-date summary of recent research outcomes in this hot topic and fulfils criteria for general interest of the Pathogens journal´ readerships. The manuscript is well prepared, very detailed and its content is adequately supported by the references.
After reviewing the manuscript, I have following comments to the authors:
Answer:
Dear reviewer,
Thanks very much for your comments!
We have tried our best to revise the manuscript. According to your comments, in the revised manuscript, we have carefully refined the Figure 1 and Table 1. Besides, we have supplemented a new Table 2 to illustrate the development of important therapeutic IL-22/22R blockers and inhibitors. In addition, the spelling mistakes have been thoroughly corrected in the revised manuscript.
We hope the revised manuscript meets with approval.
Great thanks again for your appreciation and valuable suggestion!
Best regards.
- I strongly suggest removing the text super positioning the figure object (text over the drawing) as this loses the visibility and contrast. Please, move text out but in a proximity of the figure object. This is a case of Fig 1 A-B-C-D-E.
Answer:
Thanks very much for your constructive suggestions! According to your advice, to increase the visibility and contrast of Figure 1, we have moved the text out of the figure objects in Figure 1 A-B-C-D-E and positioned it as close to the figure objects as possible. The revised figures were listed as below, and the moved text were marked by red arrows.
- There are many subjects forming the particular sections of the Figure 1. I will appreciate if the authors include internal frames selecting and separating objects of the Fig. 1A, then Fig.1B a C, Fig. D (each frame for the particular disease) and the same in Fig. 1 E for the particular bacterial infection. This will bring better orientation in the figure and will support the clarity of the figure message.
Answer:
Thanks very much for your constructive suggestions! To distinguish the particular sections in Figure 1 and improve the clarity of the figure message, we added a dashed frame in each of the Fig. 1A, Fig.1B and Fig. 1C, to include the contents in each section. Moreover, in the Fig. 1D and Fig. 1E, we added a dashed frame for each of the particular infections. The revised Figure 1 was listed above and the supplementary dashed frames were marked by red arrows.
- In Figure 1 E, correct the name of pathogens from Mycobacterium Tuberculosis to Mycobacterium tuberculosis, Citrobacter Rodentium to Citrobacter rodentium and Helicobacter Pylori to Helicobacter pylori.
Answer:
Great thanks for your careful checks! We are sorry for our carelessness and have corrected the name of pathogens from Mycobacterium Tuberculosis to Mycobacterium tuberculosis, Citrobacter Rodentium to Citrobacter rodentium and Helicobacter Pylori to Helicobacter pylori in Figure 1 E. The revised Figure 1 E was listed above and the corrected names of pathogens were marked by blue boxes.
- While in row 122 the authors use the correct style of citing “Plank et al.”, in many other cases they do not. See row 73 Trifany et al (correct is Trifany et al.), 80 Mouset et al ; 96 Lopez et al ; 113 Trifany et al ; 197 Hoffman et al ; 209 Missé et al ; 343 Mendes et al ; 365 Bunjun et al ; 536 Tripathi et al ;
Answer:
Great thanks for your careful checks! We have carefully corrected all the wrong style of citing “et al” into the correct one “et al.” in our review and the corrections were listed as below:
“Trifari et al” in row 73 has been corrected into “Trifari et al.”; “Mousset et al” in row 80 has been corrected into “Mousset et al.”; “Lopez et al” in row 96 has been corrected into “Lopez et al.”; “Trifari et al” in row 113 has been corrected into “Trifari et al.”; “Hoffmann et al” in row 175 has been corrected into “Hoffmann et al.”; “Misse´ et al” in row 205 has been corrected into “Misse´ et al.”; “Mendes et al” in row 340 has been corrected into “Mendes et al.”; “Bunjun et al” in row 365 has been corrected into “Bunjun et al.”; “Zhuang et al” in row 447 has been corrected into “Zhuang et al.”; “Tripathi et al” in row 545 has been corrected into “Tripathi et al.”.
- In chapter 2.2.2, correct “Citrobacter Rodentium” to “Citrobacter rodentium”.
Answer:
Great thanks for your careful checks! We are sorry for our carelessness. In chapter 2.2.2, We have corrected “Citrobacter Rodentium” into “Citrobacter rodentium” in row 401.
- In Table 1 in “IL-22-related treatment” column, I suggest to clearly indicate whether mentioned item is a stimulator/suppressor and should be blocked/suppressed. For instance FBXW12, IL-22BP, PstS1… It is not clear how it modulators function.
Answer:
Great thanks for your valuable comments! According to your suggestions, we have added the description of treatments (blocked/supplemented) for the target in “IL-22-related treatment” column in Table 1. Besides, we inserted a new column “Stimulatory/inhibitory target” in Table 1 to clearly illustrate the target property of each treatment. The revised Table 1 was listed below and the supplementary portions have been marked in bold red font.
Table 1. Development of IL-22-related therapies.
Infection |
IL-22-related treatment |
Stimulatory/inhibitory target |
Outcome |
References |
AIDS |
ART |
Stimulatory |
The percentage of Th22 is nearly restored. |
[181] |
Hepatitis B |
IL-22BP supplementation |
Inhibitory |
IPCs proliferation is markedly attenuated. |
[86] |
Hepatitis C |
administration of a γ-secretase inhibitor |
Inhibitory |
Both HCV-reactive Th22 cells and IL-22 levels are reduced. |
[174] |
Influenza |
silencing of FBXW12 |
Inhibitory |
The expression of IL-22R and the progression of cell cycle are both promoted in lung epithelium. |
[175] |
IL-22BP gene knockout |
Inhibitory |
The survival rate of mice infected with IAV is increased. |
[54] |
|
vunakizumab-IL22 administration |
stimulatory |
The IL-17a-mediated inflammatory responses are inhibited and the reparative capacity of IL-22 on lung tissue is elevated. |
[173] |
|
AVMC |
anti-IL-22 antibody used when IL-17A is absent |
Inhibitory |
The level of Th22 cells and the severity of AVMC are decreased, while virus replication is significantly promoted in the heart. |
[115] |
TB |
recombinant IL-22 used during MTB infection complicated with type 2 diabetes |
Stimulatory |
IL-22 controls TB by suppressing neutrophil infiltration in the alveoli, secreting elastase 2 (ELA2) and inhibiting lung epithelial cell damage. IL-22 also controls diabetes by reducing insulin and improving lipid metabolism in serum. |
[176] |
PD-1 neutralization |
Inhibitory |
The anti-PD1 antibody restores IL-22 levels and inhibits the development of tuberculosis. |
[177] |
|
PstS1 administration |
Stimulatory |
PstS1 promoted the secretion of IL-22 through activating CD8α-DCs in memory T cells. |
[178] |
|
S. pneumoniae infection |
TLR-5 agonist flagellin administration |
Stimulatory |
The activated TLR-5 pathway remodels IL-22-mediated immune responses to defend the bacterial invasion. |
[148] |
VRE infection |
oral administration of TLR-4 ligand LPS |
Stimulatory |
The activated TLR-4 pathway remodels IL-22-mediated immune responses to defend the bacterial invasion. |
[145] |
oral administration of artificial TLR-7 ligand resiquimod (R848) |
Stimulatory |
The activated TLR-7 pathway remodels IL-22-mediated immune responses to defend the bacterial invasion. |
[180] |
- In the manuscript, I miss a Table 2 summarizing the development of all important therapeutic IL-22/22R blockers. Such table should contain 3 separated parts: IL-22 cytokine inhibitors, IL-22R1 and IL-22BP blockers. These should be sorted into neutralizing antibodies/antibody fragments (Nanobody); small molecules; and small binding proteins (scaffold binders) (f.i. Affibody, Darpin, Anticalin, etc.). All presented with particular references. This would improve the practical impact of this review article.
Answer:
Great thanks for your valuable comments! We have supplemented a new Table 2 as you advised just below the Table 1. It illustrates the development of all important therapeutic IL-22/22R blockers and inhibitors, and makes our review more practical. The supplementary Table 2 was listed below:
Table 2. Development of therapeutic IL-22/22R blockers
Therapeutic IL-22/22R blockers |
Types |
References |
|
IL-22 cytokine inhibitors |
neutralizing antibodies |
anti-IL-22 antibodies |
[113, 115] |
small molecules |
T-bet, TGF-β, I-IFN, 1,25(OH)2D3, the MTB antigens CFP-10 and ESAT-6 |
[13, 20, 28, 29, 106, 177] |
|
IL-22R blockers |
small molecules |
the novel E3 ligase subunit FBXW12 |
[175] |
small binding proteins |
IL-22BP |
[51-53] |
|
IL-22BP inhibitors |
small molecules |
IL-18, prostaglandin E2 (PGE2), NLRP3 and NLRP6 inflammasomes |
[98-100] |
---------------------------------------------------------------------------------------------------------------------
We tried our best to improve the manuscript and made some changes to the manuscript. These changes will not influence the content and framework of the paper. And here we did list the changes under the “Track model” in revised paper. We appreciate for Editors/Reviewers’ warm work earnestly and hope that the correction will meet with approval. Once again, thank you very much for your comments and suggestions.
Correspondence author: Meng Zhang (zhangmeng1930@126.com) & Chaozhao Liang (liang_chaozhao@ahmu.edu.cn).
Reviewer 3 Report
GENERAL COMMENTS
The authors review the role of Th22 cells and IL22 on different models of infectious diseases, after a rational review of the definition, differentiation, and function of Th22/IL22. The manuscript is well structured y provides a relevant review (supported by numerous and appropriate bibliography references) about our current knowledge on Th22/IL22 in the context of infections. Despite this, specific recommendations for revision should be adequately addressed by the authors.
SPECIFIC RECOMMENDATIONS FOR REVISION
- Major scientific points:
1.- The authors provide many findings and data from different sources in each of the sections, but sometimes it is confusing the relationship among them (i.e., the review provides too many descriptions somehow unconnected). Therefore, it is recommended to try to associate concepts and findings, to ease reading, in each section.
2.- For some of the models of infectious diseases used, only data from mice are provided; however, the title of the study is “Current Knowledge of Th22 Cell and IL-22 Functions in Human Infectious Diseases”. Therefore, some sentences about this limitation (no data from humans) and the risk that information obtained from the mouse could be inappropriately extrapolated to the human should be added.
3. In the ms., it is stated that under different infections, Th22/IL-22 can exert a bidirectional protective or pathogenic effect on the human immune system, but it is not clear what factor(s) induce a more protective vs more deleterious role for the Th22/IL22 axis, or whether these mechanisms are not known, or just known in certain diseases, etc. This should be detailed for a more comprehensive and understandable reading.
Author Response
Point to point reply
Dear Editors and Reviewers:
Thank you for your letter and for the reviewers’ valuable comments concerning our manuscript entitled “Current Knowledge of Th22 Cell and IL-22 Functions in Infectious Diseases” (ID: pathogens-2074797). Those comments are very constructive for revising and improving our manuscript, as well as the important guiding significance to our studies. We have studied the comments carefully and have made extensive corrections which we hope meet with approval. Revised portions are marked under the “Track model” in the paper. The main corrections in the paper and the detailed point-by-point responses to the reviewer’s comments are listed as flowing:
Response to reviewers
Response to Reviewer #3:
Comments and Suggestions for Authors
GENERAL COMMENTS
The authors review the role of Th22 cells and IL22 on different models of infectious diseases, after a rational review of the definition, differentiation, and function of Th22/IL22. The manuscript is well structured y provides a relevant review (supported by numerous and appropriate bibliography references) about our current knowledge on Th22/IL22 in the context of infections. Despite this, specific recommendations for revision should be adequately addressed by the authors.
SPECIFIC RECOMMENDATIONS FOR REVISION
- Major scientific points:
Answer:
Dear reviewer,
Thanks very much for your comments!
We have tried our best to revise the manuscript. According to your comments, in each section of the revised manuscript, the association between concepts and findings has been repeatedly studied and refined. Moreover, we have changed the title of our manuscript from “Current Knowledge of Th22 Cell and IL-22 Functions in Human Infectious Diseases” into “Current Knowledge of Th22 Cell and IL-22 Functions in Infectious Diseases”, and supplemented the description about the limitation of no available data for human and the irrationality of applying the conclusions of animal experiments directly to human beings. We also supplemented the description of factors that participate in the Th22/IL22 axis and whether these factors are protective or deleterious.
We hope the revised manuscript meets with approval.
Great thanks again for your appreciation and valuable suggestion!
Best regards.
- The authors provide many findings and data from different sources in each of the sections, but sometimes it is confusing the relationship among them (i.e., the review provides too many descriptions somehow unconnected). Therefore, it is recommended to try to associate concepts and findings, to ease reading, in each section.
Answer:
Great thanks for your valuable comments! We have thoroughly checked each section of our review, and the association between concepts and findings has been repeatedly studied and refined in each section. The detailed revising parts were listed as below, and the revised parts were marked in red. Moreover, the adapted concepts and findings are highlighted in yellow and blue respectively.
(1). 2.1.1 COVID-19
Before revision:
COVID-19 is a pneumonia induced by severe acute respiratory syndrome corona-virus 2 (SARS-CoV-2). Researchers in infectious diseases have focused on the patho-genic mechanisms of COVID-19 since its outbreak in 2019. In the acute phase of COVID-19, immune responses cause dramatic increases in multiple cytokines, including IL-22, which is mainly produced by Th22 cells [56, 57]. This process is called cytokine release syndrome (CRS). Importantly, fatal complications of COVID-19, such as multi-ple organ failure and acute respiratory distress syndrome, have been demonstrated to be correlated with CRS [58, 59]. SARS-CoV-2 can also lead to a multisystem inflammatory syndrome in children (MIS-C). Serum IL-22 levels are both elevated in children with COVID-19 and those with MIS-C [60]. COVID-19 patients with different prognoses have similar IL-22 levels, suggesting that IL-22 does not affect the outcomes of SARS-CoV-2 infection [60]. Nevertheless, studies have discovered that in patients with fulminant COVID-19-related myocarditis, some met the criteria for multisystem inflammatory syndrome (MIS-A+), whereas the rest did not (MIS-A-) [61]. Compared to the MIS-A- group, patients who were MIS-A+ showed a higher expression of IL-22 as well as a better prognosis and lower mortality [61]. These results suggested that IL-22 may be protective and antiviral in MIS-A+ COVID-19 patients. Notably, a novel study pointed out that abnormal dynamic IL-22R1 expression on blood myeloid cells and CD4+ T cells is a characteristic manifestation of SARS-CoV-2 infection [62]. IL-22R1 expression on mye-loid cells varies according to the severity of the disease, which may be useful in dis-criminating the course of the disease [62]. Furthermore, the number of IL-22R1-expressing myeloid cells is correlated with increased concentrations of COVID-19-related immune mediators [62]. This result indicated that IL-22R1 may par-ticipate in the cascade leading to CRS. Studies have also proposed that the IL-22-induced signaling pathway may switch from protective to pathogenic as the disease progresses [62]. Hoffmann et al proposed that COVID-19 shares some similar symptoms with in-fluenza and respiratory syncytial virus (RSV)-induced pneumonia [63]. According to previous studies, IL-22/Th22 is protective against influenza and RSV pneumonia [54, 64, 65]. Therefore, IL-22/Th22 may also play a critical role in the pathological process of COVID-19, and its effect still awaits further research.
After revision:
COVID-19 is pneumonia induced by severe acute respiratory syndrome corona-virus 2 (SARS-CoV-2). Studies in the context of infectious diseases have focused on the pathogenic mechanisms of COVID-19 since its outbreak in 2019. Hoffmann et al. proposed that COVID-19 shares some similar symptoms with influenza and respiratory syncytial virus (RSV)-induced pneumonia [56]. According to previous studies, IL-22/Th22 is protective against influenza and RSV pneumonia [54, 57, 58] and may exert a similar effect on COVID-19. In patients with fulminant COVID-19-related myocarditis, some met the criteria for multisystem inflammatory syndrome (MIS-A+), whereas the rest did not (MIS-A-) [59]. Compared to the MIS-A- group, MIS-A+ patients showed a higher expression of IL-22 as well as a better prognosis and lower mortality [59]. These results suggest that IL-22 may have a protective and antiviral effect in MIS-A+ COVID-19 patients. A novel study noted that abnormal dynamic IL-22R1 expression on blood myeloid cells and CD4+ T cells is a characteristic of SARS-CoV-2 infection [60]. The IL-22R1 expression on myeloid cells is discriminative for the severity of COVID-19 [60]. However, COVID-19 patients with different prognoses have similar IL-22 levels, suggesting that IL-22 does not affect the outcomes of SARS-CoV-2 infection [61]. Furthermore, the number of IL-22R1-expressing myeloid cells is correlated with the plasma levels of COVID-19-related immune mediators [60]. During the acute phase of COVID-19, the immune response leads to a dramatic increase in several cytokines, including IL-22, which is mainly produced by Th22 cells [62, 63]. This process is called cytokine release syndrome (CRS) and contributes to fatal com-plications, such as multiple organ failure and acute respiratory distress syndrome [64, 65]. It indicated that IL-22R1+ myeloid cells may participate in the cascade leading to CRS and promoting the deterioration of COVID-19. This study also suggested that the IL-22-induced signaling pathway switches from protective to pathogenic as the disease progresses [60]. Therefore, IL-22/Th22 cells may play a critical role in the pathological process of COVID-19, but the detailed mechanism still awaits further research.
(2). 2.1.2 AIDS
Before revision:
Acquired immunodeficiency syndrome (AIDS), an infectious illness that arises from infection with human immunodeficiency virus (HIV), is known for its high mor-tality and the prolonged course of disease [66]. In comparison with healthy controls and HIV-infected patients, more acute-phase serum amyloid A (A-SAA) and IL-22 are produced in HIV-exposed but uninfected individuals (EUs) [67, 68]. Since IL-22 can promote the expression of acute phase proteins such as A-SAA and β-defensins 2 and 3 in epithelial cells and liver cells [47, 49, 69], Misse´ et al speculated that in EUs, IL-22 enhances body resistance to HIV infection in an acute phase protein-related manner [67]. They coincubated A-SAA with immature DCs in vitro and discovered that A-SAA is an agonist of formyl peptide receptor (FPR) and can enhance FPR expression on DCs. The FPR then promoted the phosphorylation of CCR5 and decreased the expression of CCR5 on DCs, resulting in a decreased susceptibility of DCs to HIV [67].
After revision:
Acquired immunodeficiency syndrome (AIDS), an infectious illness that arises from infection with human immunodeficiency virus (HIV), is known for its high mortality and prolonged course [66]. In comparison with healthy controls and HIV-infected patients, more acute-phase serum amyloid A (A-SAA) and IL-22 are produced in HIV-exposed but uninfected individuals (EUs) [67, 68]. IL-22 has been confirmed to promote the expression of A-SAA in epithelial cells and liver cells [47, 49, 69]. Moreover, Misse et al. co-incubated A-SAA with immature DCs in vitro for further exploration. It discovered that A-SAA is an agonist of formyl peptide receptor (FPR) and enhances the FPR expression on DCs [67]. The FPR then promoted the phosphorylation of CCR5 and decreased the expression of CCR5 on DCs, resulting in a decreased susceptibility of DCs to HIV [67]. Therefore, it was suggested that the high resistance of EUs to HIV may be associated with IL-22-induced A-SAA.
(3). 2.1.5 Acute Viral Myocarditis
Before revision:
Acute viral myocarditis (AVMC) is a nonspecific interstitial myocardial inflam-mation, with coxsackievirus B3 (CVB3) infection being its major cause [109]. Without timely treatment, AVMC can progress to dilated cardiomyopathy (DCM) [109]. Com-pared to the controls, circulating Th22 cells and IL-22 were significantly upregulated in mice infected with CVB3 [110]. In addition, administration of an anti-IL-22 antibody in CVB3-infected mice upregulated the levels of proinflammatory cytokines such as IL-17, IL-6 and TNF-α [110]. It also decreased the frequency of antiviral IFN-γ and resulted in deterioration of AVMC, demonstrating that Th22/IL-22 may have an antiviral and prognosis-improving effect during CVB3 infection by regulating the expression of cy-tokines. Using animal models of CVB3-induced chronic myocarditis and DCM, studies revealed that Th22/IL-22 is also protective in chronic myocarditis and that IL-22 can in-hibit cardiac fibrosis [111]. Therefore, Th22/IL-22 may be a promising target for treating coxsackievirus-induced acute viral myocarditis, chronic myocarditis, and DCM. Nev-ertheless, in IL-17A-/- mice with AVMC, IL-22 neutralization contributes to the im-provement of acute myocarditis but increases viral replication at the same time [112]. This result suggested that when IL-17A is absent, IL-22 can exacerbate the progression of AVMC and inhibit CVB3 replication。
After revision:
Acute viral myocarditis (AVMC) is a nonspecific interstitial myocardial inflammation, with coxsackievirus B3 (CVB3) infection being the major cause [112]. Compared to the controls, circulating Th22 cells and IL-22 were significantly upregulated in mice infected with CVB3 [113]. In addition, administration of anti-IL-22 antibody in CVB3-infected mice decreased the frequency of antiviral IFN-γ and upregulated the levels of proinflammatory cytokines such as IL-17, IL-6 and TNF-α [113]. Consequently, the antibody resulted in the deterioration of AVMC, demonstrating that Th22/IL-22 can regulate the expression of cytokines and improve antiviral and prognosis during CVB3 infection. If left untreated, AVMC can progress to dilated cardiomyopathy (DCM) [112]. Using animal models of CVB3-induced chronic myocarditis and DCM, studies have revealed that Th22/IL-22 also has a protective role in chronic myocarditis and that IL-22 can inhibit cardiac fibrosis [114]. Therefore, Th22/IL-22 may be a promising target for treating CVB3-induced acute viral myocarditis, chronic myocarditis, and DCM. Nevertheless, in IL-17A-/- mice with AVMC, IL-22 neutralization contributes to the improvement of acute myocarditis but increases viral replication at the same time [115]. This result suggested that when IL-17A is absent, IL-22 can exacerbate the progression of AVMC and inhibit CVB3 replication.
(4) 2.2.1 Mycobacterium tuberculosis
Before revision:
Mycobacterium tuberculosis (MTB) is a common pathogen that is primarily transmitted via the respiratory tract. After invading the human body, it can cause asymptomatic MTB-specific immune responses, latent tuberculosis (TB), or active TB [125]. Th22 cells are the main IL-22 producer during MTB infection, while other subsets, such as Th1 cells, CD8+ T cells, and NKT cells, also secrete IL-22 [126-128]. Compared to healthy controls, TB patients had fewer IL-22 and IL-22+ T cells in plasma [129, 130]. In addition, the bronchoalveolar lavage fluid (BALF) of pulmonary TB patients contains a large amount of IL-22, and its level is significantly higher than that in the corresponding plasma [131]. IL-22 was also found to be abundant in both TB-induced pleural and per-icardial effusions [132]. Therefore, it illustrated a possible clustering of IL-22-producing cells in disease sites of TB patients. Experiments have proven that MTB in the lungs and pulmonary tuberculosis granulomas accumulate IL-22+ T cells [133]. Furthermore, in tuberculous pleurisy, the accumulation of Th22 at the disease site was associated with the chemotactic effect of cytokines in tuberculous pleural effusion (TPE) as well as the pleural mesothelial cell (PMC)-expressed chemokines CCL20, CCL22 and CCL27 [134]. PMCs also promote Th22 proliferation and differentiation by presenting MTB antigens [134]. Bunjun et al observed a high level of IL-22 in latent MTB patients stimulated by MTB antigens, and IL-22 takes up the greatest proportion of the responsive CD4+ re-sponses [126]. During the chronic phase of MTB infection, IL-22-/- mice showed a higher susceptibility to MTB HN878 and a higher bacterial load in the lungs than the wild-type controls [127]. In addition, the pulmonary MTB load at the early stage is significantly increased in IL-22-deficient mice [135]. Therefore, IL-22 is required in both adaptive and innate immune responses against MTB.
As research deepens, the pulmonary recruitment of macrophages has been demonstrated to be IL-22-dependent [127]. IL-22 stimulates macrophage infiltration by enhancing CCL2 expression on epithelial cells [127]. Furthermore, upon stimulation by MTB, Th22 cells can evolve into membrane-bound IL-22+ (mIL-22+) Th22 cells to extend the half-life of IL-22 [136]. During MTB infection, IL-22R is mainly expressed on the surface of macrophages that accumulate within tuberculous granulomas [127]. It may contribute to mIL-22+ Th22 cell accumulation at disease sites [136]. More importantly, after binding to IL-22R on infected macrophages, mIL-22 can inhibit intracellular MTB replication [136]. In MTB-infected phagocytes, IL-22 modulates the expression of Rab7 and Rab14 by upregulating the production of calgranulin A [137, 138]. The regulation of Rab7 and Rab14 then inhibits intracellular MTB replication and promotes phagosomal maturation and fusion [137, 138]. Furthermore, TNF-α can directly promote macrophage activation and MTB control, and IL-22 was found to stimulate TNF-α production by IL-22R+ macrophages [127]. …
After revision:
Mycobacterium tuberculosis (MTB) is a common pathogen that is primarily transmitted via the respiratory tract [128]. Compared to healthy controls, tuberculosis (TB) patients had fewer IL-22 and IL-22+ T cells in plasma [129, 130]. Th22 cells are the main IL-22 producer during MTB infection, while other subsets, such as Th1 cells, CD8+ T cells, and NKT cells, also secrete IL-22 [131-133]. In addition, the bronchoalveolar lavage fluid (BALF) of pulmonary TB patients contains a large amount of IL-22, at significantly higher levels than the corresponding plasma [134]. IL-22 was also found to be abundant in both TB-induced pleural and pericardial effusions [135]. Therefore, it illustrated a possible aggregation of IL-22-producing cells in disease sites of TB patients. MTB in TB granulomas and pulmonary TB granulomas have been shown to recruit IL-22+ T cells [136]. During MTB infection, since IL-22R is mainly expressed on the surface of macrophages in tuberculous granulomas, it may also contribute to Th22 cell accumulation [132, 137]. Furthermore, in tuberculous pleurisy, the accumulation of Th22 cells at the disease site was associated with the chemotactic effect of cytokines in tuberculous pleural effusion (TPE) as well as the pleural mesothelial cell (PMC)-expressed chemokines CCL20, CCL22 and CCL27 [138]. PMCs also promote Th22 proliferation and differentiation by presenting MTB antigens [138]. MTB invasion human body can result in asymptomatic specific immune responses, latent TB, or active TB [128]. Bunjun et al. observed a high level of IL-22 in latent MTB patients stimulated by MTB antigens, and IL-22 accounted for the largest proportion of the responsive CD4+ responses [131]. During the chronic phase of MTB infection, IL-22-/- mice showed a higher susceptibility to MTB HN878 and a higher bacterial load in the lungs than wild-type controls [132]. In addition, the pulmonary MTB load at the early stage is significantly increased in IL-22-deficient mice [139]. Therefore, IL-22 is required in both adaptive and innate immune responses against MTB.
Research has shown that in response to stimulation by MTB, Th22 cells can evolve into membrane-bound IL-22+ (mIL-22+) Th22 cells to extend the half-life of IL-22 [137]. More importantly, mIL-22 binds to IL-22R on infected macrophages to inhibit intracellular MTB replication [137]. IL-22 also enhances the expression of CCL2 on epithelial cells to stimulate the pulmonary recruitment of macrophages [132]. In MTB-infected phagocytes, IL-22 modulates the expression of Rab7 and Rab14 by up-regulating the production of calgranulin A [140, 141]. Rab7 and Rab14 subsequently inhibit intracellular MTB replication and promote phagosomal maturation and fusion [140, 141]. Furthermore, IL-22 was found to stimulate TNF-α production by IL-22R+ macrophages [132]. TNF-α can directly promote macrophage activation and MTB control. …
- For some of the models of infectious diseases used, only data from mice are provided; however, the title of the study is “Current Knowledge of Th22 Cell and IL-22 Functions in Human Infectious Diseases”. Therefore, some sentences about this limitation (no data from humans) and the risk that information obtained from the mouse could be inappropriately extrapolated to the human should be added.
Answer:
Great thanks for your valuable comments! We have changed the title of this review from “Current Knowledge of Th22 Cell and IL-22 Functions in Human Infectious Diseases” into “Current Knowledge of Th22 Cell and IL-22 Functions in Infectious Diseases”. Moreover, in line 589, we supplemented the description about the limitation of no available data for human and the irrationality of applying the conclusions of animal experiments directly to human beings. The detailed supplementary part was listed below:
Nevertheless, one limitation of the studies we have compiled in this review is that some results regarding the role of Th22/IL-22 in bacterial and viral infections have only been confirmed in mouse experiments, with no available human experimental data. Since it is unreasonable to apply the conclusions of animal experiments directly to human beings, there is currently very limited knowledge about how Th22/IL-22 plays a role in human infectious diseases.
- In the ms., it is stated that under different infections, Th22/IL-22 can exert a bidirectional protective or pathogenic effect on the human immune system, but it is not clear what factor(s) induce a more protective vs more deleterious role for the Th22/IL22 axis, or whether these mechanisms are not known, or just known in certain diseases, etc. This should be detailed for a more comprehensive and understandable reading.
Answer:
Great thanks for your valuable comments! We collated the factors that participate in the Th22/IL22 axis and whether these factors are protective or deleterious. The corresponding description has been supplemented in line 506-515 of the article, making the mechanism of Th22/IL22 axis more comprehensive. The detailed supplementary part was listed below:
Based on the above, cytokines such as TNF-α, IFN-λ and IL-23, transcription factors such as T-bet and AhR, and chemokines such as CCL2, CCL17 and CXCL13 have been shown to have a protective role in the Th22/IL-22 axis. Other effector molecules, including C3, MMP-9, Rab7, Rab14, the TLR-5 agonist flagellin, and the antibacterial proteins RegIIIβ and RegIIIγ, are also involved in IL-22-induced anti-infectious immunity. Moreover, the matrix metalloproteinase MMP-10, the antimicrobial protein S100A9 and chemokines including CXCL9, CXCL10, CXCL2, and CXCL16 are risk factors for the Th22/IL-22 downstream signaling system. Interestingly, STAT3 and the antimicrobial protein cal-granulin A can be both protective and deleterious in the Th22/IL-22 axis, depending on the type of infectious disease.
---------------------------------------------------------------------------------------------------------------------
We tried our best to improve the manuscript and made some changes to the manuscript. These changes will not influence the content and framework of the paper. And here we did list the changes under the “Track model” in revised paper. We appreciate for Editors/Reviewers’ warm work earnestly and hope that the correction will meet with approval. Once again, thank you very much for your comments and suggestions.
Correspondence author: Meng Zhang (zhangmeng1930@126.com) & Chaozhao Liang (liang_chaozhao@ahmu.edu.cn).
Round 2
Reviewer 3 Report
The ms is now significantly improved, after addressing the recommendation of my (and other Reviewers) suggestions.
Author Response
Response letter
Dear Editors and Reviewers:
Thank you for your letter and for the reviewer’s valuable comments concerning our manuscript entitled “Current Knowledge of Th22 Cell and IL-22 Functions in Infectious Diseases” (ID: pathogens-2074797). Those comments are very constructive for revising and improving our manuscript, as well as the important guiding significance to our studies. We have studied the comments carefully and have made extensive corrections which we hope meet with approval. Revised portions are marked under the “Track model” in the paper. The main corrections in the paper and the detailed point-by-point responses to the reviewer’s comments are listed as flowing:
Response to reviewers
Response to Reviewer # 3:
Comments and Suggestions for Authors
The ms is now significantly improved, after addressing the recommendation of my (and other Reviewers) suggestions.
Answer:
Dear reviewer,
Thanks very much for your comments!
We have tried our best to revise the manuscript. According to your comments, the grammar and spell mistakes have been thoroughly corrected in the revised manuscript. Besides, we have scrutinized the language of this article. To enhance the precision and scientific quality of our review, we have checked the corresponding references and made modifications to the descriptions of some conclusions. Following your comments, our manuscript became more concise and easier to understand.
We hope the revised manuscript meets with approval.
Great thanks again for your appreciation and valuable suggestions!
Best regards.
Answer:
Great thanks for your valuable suggestion! According to your advice, we have thoroughly checked the manuscript and carefully corrected the grammar and spell faults. The revised parts were marked in the manuscript under “Track model”, and some of them were listed as below:
1.1 Grammar revision:
- Before the revising:
Then, the interactions of IL-10R2 and the IL-22/IL-22R1 complex form an IL-22/IL-22R1/IL-10R2 triad and activate downstream signaling pathways [33].
After the revising:
Then, the interactions between IL-10R2 and the IL-22/IL-22R1 complex form an IL-22/IL-22R1/IL-10R2 triad and activate downstream signaling pathways [33].
- Before the revising:
Research advances in Th22-targeted therapies in infection will also be discussed.
After the revising:
Research advances in Th22-targeted therapies for infection will also be discussed.
- Before the revising:
Moreover, Misse´ et al. co-incubated A-SAA with immature DCs in vitro for further exploration. It discovered that A-SAA is an agonist of formyl peptide receptor (FPR) and enhances the FPR expression on DCs [67].
After the revising:
Moreover, Misse´ et al. coincubated A-SAA with immature DCs in vitro for further exploration. A-SAA is an agonist of the formyl peptide receptor (FPR) and enhances FPR expression on DCs [67].
- Before the revising:
In addition, administration of anti-IL-22 antibody in CVB3-infected mice decreased the frequency of antiviral IFN-γ and upregulated the levels of proinflammatory cytokines such as IL-17, IL-6 and TNF-α [113].
After the revising:
Furthermore, in CVB3-infected mice, the anti-IL-22 antibody reduced the antiviral IFN-γ production while increasing the levels of proinflammatory cytokines such as IL-17, IL-6 and TNF-α [113].
- Before the revising:
IL-22 also induces the pulmonary production of IFN-λ, thereby blocking the release of inflammatory mediators such as IL-1β [164].
After the revising:
IL-22 also induces pulmonary IFN-λ production, preventing the release of inflammatory mediators such as IL-1β [164].
1.2 Spell revision:
“Th22 Cells” in row 52 has been corrected into “Th22 cells”; “Naive” in row 53 and 54 has been corrected into “Naïve”; “Tbet” in Figure 1 (B) and row 57 has been corrected into “T-bet”; “CCR4, coexpresses C-C motif chemokine receptor 4” in row 63 has been corrected into “CCR4, C-C motif chemokine receptor 4”; “secreted” in row 73 has been corrected into “secrete”; “producer” in row 77 and 350 has been corrected into “producers”; “suggested” in row 224 has been corrected into “suggests”; “phagosomal” in row 378 has been corrected into “phagosome”; “Reg3γ” in row 387 has been corrected into “RegIIIγ”; “bending” in row 434 has been corrected into “binding”; “prolongedsurvival” in row 436 has been corrected into “prolonged survival”; “are” in row 445 has been corrected into “is”; “leads” in row 489 has been corrected into “lead”; “infection” in row 498 has been corrected into “infections”; “IL-17a” in row 531 and Table 1 (row 7 and column 4) has been corrected into “IL-17A”; “REGIIIγ” in row 558 and 561 has been corrected into “RegIIIγ”.
Answer:
Great thanks for your valuable suggestion! According to your advice, the language in this manuscript has been carefully checked and refined for higher precision and scientific quality. The revised parts were listed as below:
(1)Before the revising:
However, an in vitro study confirmed that two variants of the gene encoding IL-22BP are correlated with HCV-mediated liver fibrosis and cirrhosis, and that culture with high levels of IL-22 resulted in a protective immune response to hepatitis C [51].
After the revising:
However, an in vitro study confirmed that two variants of the gene encoding IL-22BP are correlated with HCV-mediated liver fibrosis and cirrhosis. It has aslo been proven that high production of IL-22 is correlated with protective immune responses to hepatitis C [51].
(2)Before the revising:
Consequently, the antibody resulted in the deterioration of AVMC, demonstrating that Th22/IL-22 can regulate the expression of cytokines and improve antiviral and prognosis during CVB3 infection.
After the revising:
Consequently, the antibody resulted in the deterioration of AVMC, demonstrating that Th22/IL-22 can regulate the expression of cytokines and improve antiviral activity and prognosis during CVB3 infection.
(3)Before the revising:
Using animal models of CVB3-induced chronic myocarditis and DCM, studies have revealed that Th22/IL-22 also has a protective role in chronic myocarditis and that IL-22 can inhibit cardiac fibrosis [114]. Therefore, Th22/IL-22 may be a promising target for treating coxsackievirus-induced acute viral myocarditis, chronic myocarditis, and DCM.
After the revising:
Using animal models of CVB3-induced chronic myocarditis and DCM, studies have revealed that Th22/IL-22 also has a protective role in chronic viral myocarditis and that IL-22 can inhibit cardiac fibrosis [114]. Therefore, Th22/IL-22 may be a promising target for treating coxsackievirus-induced acute viral myocarditis, chronic viral myocarditis, and DCM.
(4)Before the revising:
Hand, foot, and mouth disease (HFMD) occurs mainly in children infected with coxsackievirus-A16 (CV-A16) or human enterovirus-71 (EV-71), leading to characteristic herpes on the hand, foot, mouth, and buttock [116].
After the revising:
Hand, foot, and mouth disease (HFMD) occurs mainly in children infected with coxsackievirus A16 (CV-A16) or enterovirus 71 (EV-71), leading to character-istic herpes on the hand, foot, mouth, and buttock [116].
(5)Before the revising:
MTB in TB granulomas and pulmonary TB granulomas have been shown to recruit IL-22+ T cells [136].
After the revising:
MTBs in pulmonary TB granulomas have been shown to recruit IL-22+ T cells [136].
---------------------------------------------------------------------------------------------------------------------
We tried our best to improve the manuscript and made some changes to the manuscript. These changes will not influence the content and framework of the paper. And here we did list the changes under the “Track model” in revised paper. We appreciate for Editors/Reviewers’ warm work earnestly and hope that the correction will meet with approval. Once again, thank you very much for your comments and suggestions.
Correspondence author: Meng Zhang (zhangmeng1930@126.com) & Chaozhao Liang (liang_chaozhao@ahmu.edu.cn).